

# DYNAMICAL PROPERTIES AND EXTREMES OF NORTHERN HEMISPHERE CLIMATE FIELDS OVER THE PAST 60 YEARS

Davide Faranda[1], Gabriele Messori[2,3], M. Carmen Alvarez-Castro[1], and Pascal Yiou[1]

[1]LSCE-IPSL, CEA Saclay l'Orme des Merisiers, CNRS UMR 8212 CEA-CNRS-UVSQ, Université Paris-Saclay, 91191 Gif-sur-Yvette, France
[2]Department of Meteorology, Stockholm University and Bolin Centre for Climate Science, Stockholm, Sweden
[3]Currently: visiting researcher at LSCE-IPSL, CEA Saclay l'Orme des Merisiers, CNRS UMR 8212 CEA-CNRS-UVSQ, Université Paris-Saclay, 91191 Gif-sur-Yvette, France

*Correspondence to:* D Faranda (davide.faranda@cea.fr)

**Abstract.** Atmospheric dynamics is described by a set of partial differential equation yielding an infinite dimensional phase space. However, the actual trajectories followed by the system appear to be constrained to a finite dimensional phase space, i.e. a strange attractor. The dynamical properties of this attractor are difficult to determine due to the complex nature of atmospheric motions. A first step to simplify the problem is to focus on observables which affect – or are linked to phenomena which affect – human welfare and activities, such as sea level pressure, near-surface temperature and precipitation. We make use of recent advances in dynamical systems theory to estimate two instantaneous dynamical properties of the above fields for the Northern Hemisphere: local dimension and persistence. We then use these metrics to characterise the seasonality of the different fields and their interplay. We further analyse the large-scale anomaly patterns corresponding to phase-space extremes – namely timesteps at which the fields display extremes in their instantaneous dynamical properties. The analysis is based on the NCEP/NCAR reanalysis data, over the period 1948-2013. The results show that: i) despite the high dimensionality of atmospheric dynamics, the Northern Hemisphere sea level pressure and temperature fields can on average be described by roughly twenty degrees of freedom; ii) the precipitation field has a higher dimensionality; iii) the seasonal forcing modulates the variability of the dynamical indicators and affects the occurrence of phase-space extremes. We further identify a number of robust correlations between the dynamical properties of the different variables.



# 1  Introduction

Atmospheric motions are governed by a web of complex interactions among the different components of the earth system (Charney, 1947). Solar radiation and the earth's rotation are the primary large-scale drivers of the chaotic atmospheric dynamics, while turbulent motions add a layer of complexity at small scales. This picture is further complicated by the presence of features such as ocean-land interactions, vegetation, human-induced forcing and the hydrological cycle. Understanding both the transient (i.e. meteorological) and mean (i.e. climatic) properties of this system is one of today's major scientific challenges.

Since Lorenz (1963)'s seminal work, dynamical systems techniques have been widely applied to the study of the atmosphere. For example, the use of tools such as the Lyapunov exponents or the Kolmogorov-Sinai entropy have led to important advances in our understanding of atmospheric predictability (Zeng et al., 1993). An important result has been to show that atmospheric motions are chaotic but not completely random: their trajectories stay close to a high dimensional object called the attractor (Lorenz, 1969; Carrassi et al., 2008; Ghil et al., 2008; Vannitsem, 2014). This object occupies only a fraction of the atmospheric phase space, meaning that its dimension $D$ is smaller than the number of variables used to describe the system. $D$ is important because it represents the number of degrees of freedom of the system, namely the minimum number of variables needed to represent the dynamics. The computation of $D$ for atmospheric attractors has posed a challenged to the dynamical systems community for several decades. Whereas in the early '80s several estimates pointed to a low dimensional $D < 10$ attractor (Fraedrich, 1986), a later review of the numerical limitations of the available techniques suggested that they tended to underestimate $D$ for complex systems Lorenz (1991). However, further estimates of $D$ were hardly attempted, because $D > 10$ implies that low dimensional models should fail in describing the atmospheric dynamics.

$D$ is a mean property, since it describes the dimension of the attractor for the whole atmospheric trajectory. However, it is often more useful to determine instantaneous dynamical systems metrics that describe transient states $\zeta$ of complex attractors. A quantity that contains such information is the local dimension $d(\zeta)$ (Lucarini et al., 2016). The value of $d$ is proportional to the active number of degrees of freedom and provides information on how predictable the state $\zeta$ and its future evolution are (Faranda et al., 2017). By averaging $d$ over all possible $\zeta$, one recovers the attractor dimension $D$. Unfortunately, the computation of $d$ traditionally posed even greater challenges than that of $D$. The original method developed by Liebovitch and Toth (1989) used box counting techniques. First, a small portion of the phase space is partitioned in hypercubes of different sizes. Then one looks at the amount of space filled up in each hypercube. The scaling of this quantity across different scales is proportional to $d$. The complexity of this technique prevented computations for high dimensional systems such as atmospheric flows. Very recently, some of the authors of this paper have contributed to develop an alternative way to obtain $d$, based on the universal behavior of Poincaré recurrences in chaotic systems (Freitas et al., 2010; Faranda et al., 2011, 2013). In few words (see Methodology and Data for the details), the recurrences of a state $\zeta$ of a chaotic dynamical system of arbitrary dimension have a universal distribution, provided that one observes enough recurrences. The parameters of this distribution are linked to the instantaneous dimension $d(\zeta)$ and to another important dynamical quantity. the inverse of the average persistence time of



the trajectory around $\theta(\zeta)$ (Freitas et al., 2012). Estimating these parameters via Poincaré recurrences is easier than with the box counting algorithms because the method avoids altogether computations in scale space.

Having overcome the technical difficulties inherent to the calculation of instantaneous dynamical systems metrics, the re-
maining step is the choice of the states $\zeta$ of interest. Since it is unpractical – not to say impossible – to consider all atmospheric observables at once, we focus our analysis on variables which are representative of events which affect humans, namely: sea level pressure (slp; cyclones, windstorms etc.), near-surface temperature (air; heat waves, cold spells) and precipitation rate (prp; droughts, floods). In dynamical systems terms, these fields represent a reduction of the original dynamics to peculiar subspaces, and are therefore special Poincaré sections. In Faranda et al. (2016) and Messori et al. (2017) we have shown that $d$
and $\theta$ can be used to characterise large-scale atmospheric fields and further provide information on the predictability linked to a given atmospheric state. It is therefore important to investigate these indicators at different spatial scales to fully understand the insights they can provide. In this study we present a novel analysis for the whole Northern Hemisphere (NH) based on $d$ and $\theta$. We further investigate for the first time the mutual correlations between the dynamical properties of different climate variables.

The paper is organised as follows: in section 2 we give an overview of the dynamical indicators and of the methodology used to compute them as well as the data used. In section 3 we present and discuss the dynamical properties of each of the three atmospheric fields separately, while in section 4 we analyse them jointly. Finally, we discuss our results and summarise our conclusions in section 5.

## 2 Methodology and Data

The attractor of a dynamical system is a geometrical object defined in the space hosting all the possible states of a system (the so-called phase space) (Milnor, 1985). Each point on the attractor $\zeta$ can be characterized by two dynamical quantities: i) the local dimension $d(\zeta)$, which provides the number of degrees of freedom active locally around $\zeta$ and ii) the inverse persistence of the state $\zeta$: $\theta(\zeta)$, which is a measure of the mean residence time of the system around $\zeta$.

### 2.1 Local dimensions

The term *attractor dimension* usually refers to a global measure (hereafter $D$) (Grassberger and Procaccia, 1983). $D$ indicates the average number of degrees of freedoms of a dynamical system. Several methods to measure $D$ were developed in the '80s (Grassberger and Procaccia, 1984; Halsey et al., 1986). These techniques have a certain number of adjustable parameters and require the system to be embedded in a subspace of the phase space. They provide good estimates of $D$ only when the trajecto-
ries are sufficiently long to estimate the embedding parameters. Such computations are therefore problematic in systems with large numbers of degrees of freedom and give biased results when applied to atmospheric flows (Grassberger, 1986; Lorenz,



1991).

The technique we exploit results from the application of extreme value theory to Poincaré recurrences in dynamical systems (Freitas et al., 2010; Faranda et al., 2011). In this approach, the returns to points on chaotic attractors are fully characterised by extreme value laws. In practice, one needs long trajectories $x(t)$ that approximate sequences of states on the attractor. One then fixes a point $\zeta$ of the trajectory and computes the probability $P$ that $x(t)$ returns in a ball of radius $\epsilon$ centered on the point $\zeta$. The Freitas et al. (2010) theorem, modified in Lucarini et al. (2012), states that logarithmic returns: $g(x(t)) = -\log(dist(x(t), \zeta))$ are distributed as:

$$P(g(x(t)) > s(q), \zeta) \simeq \exp\left[-\frac{x - \mu(\zeta)}{\sigma(\zeta)}\right]$$

Here $s$ is a high threshold associated to a quantile $q$, linked to the radius $\epsilon$ via $s = g^{-1}(\epsilon)$. In other words, requiring that the orbit falls within a ball of radius $\epsilon$ around the point $\zeta$ is equivalent to asking that the series $g(x(t))$ is over the threshold $s$. The resulting distribution is simply the exponential member of the Generalized Pareto Distribution family. The parameters $\mu$ and $\sigma$ depend on the point $\zeta$ chosen on the attractor. $\sigma(\zeta)$ then provides the local dimensions $d(\zeta)$ via the simple relation $\sigma = 1/d(\zeta)$. This result is very powerful because it provides a new way to compute local dimensions on the attractor and to recover $D$ as the average of $d$ on all the $\zeta$s without the need for embedding. We want to stress that this procedure is not just a statistical fitting. The reason why it provides good estimates of $d$ that were impossible to obtain with previous techniques derives from the universality of the extreme value statistics for Poincaré recurrences: one knows a priori the statistics of such recurrences and can then check that they are achieved for the numerical trajectory examined.

## 2.2 Local persistence

The stability of the state $\zeta$ is measured by $\theta(\zeta)$: namely the inverse of the average residence time of trajectories around $\zeta$. For discrete maps $\theta$ is uniquely defined (see Freitas et al. (2012) for details): if $\zeta$ is a fixed point of the dynamics, $\theta(\zeta) = 0$. For a point that leaves the neighborhood of $\zeta$ immediately, $\theta = 1$. For continuous flows, the definition of $\theta$ depends on the Poincaré map chosen and precisely on the $\Delta t$ chosen to descritize the flow. Since $\theta$ is the inverse of the average residence time, its unit measure will be $1/\Delta t$. In general, the higher the persistence of the point $\zeta$, the longer the previous and subsequent states of the system will resemble $\zeta$. The residence time can be computed by introducing a further parameter in the previous law. This parameter, known as extremal index is such that:

$$P(g(x(t)) > q) \simeq \exp\left[-\theta\left(\frac{x - \mu(\zeta)}{\sigma(\zeta)}\right)\right]$$

To estimate $\theta$, we adopt the Süveges' estimator (Süveges, 2007). For a fixed quantile $q$, the estimator reads:

$$\theta = \frac{\sum_i^{N_c}(1-q)S_i + N + N_c - \sqrt{\left(\sum_i^{N_c}(1-q)S_i + N + N_c\right)^2 - 8Nc\sum_i^{N_c}(1-q)S_i}}{2\sum_i^{N_c}(1-q)S_i},$$





where $N$ is the number of recurrences above the chosen quantile, $N_c$ the number of observations which form a cluster of at least two consecutive recurrences and $S_i$ the length of each cluster $i$. For details of the derivation of this estimator, the reader is referred to Süveges (2007).

## 2.3 Data

We use daily fields from the NCEP/NCAR reanalysis (Kalnay et al., 1996), with a horizontal resolution of $2°$. The analysis is carried out over the whole Northern Hemisphere for all days of the year over the period 1948–2013. The observables of interest are sea-level pressure, 1000 hPa temperature and precipitation rate. A previous study (Faranda et al., 2017) has shown that the results obtained are largely independent of the dataset used and of its spatial resolution.

Anomalies are defined as deviations from the long-term daily mean. So, for example, the anomaly of air at a given location on the $5^{th}$ December 2000 is computed relative to the mean value of all $5^{th}$ Decembers in the dataset at that location.

## 3    Dynamical properties of individual observables

## 3.1    Sea-Level Pressure

The local dimension, $d$, of the slp field shows a marked variability throughout the analysis period, with values ranging from as
low as 8.6 to as high as 33.2 (Figure 1a). The average dimension $D$, which in this case is roughly 19.4, therefore provides a very incomplete information concerning the field of interest, since the number of locally active degrees of freedom (identified by $d$) can vary by a factor of almost 4. The autocorrelation function (ACF) of $d$ (Figure 1c) highlights a robust variability pattern which is not immediately evident from the raw time series. A previous analysis has highlighted the strong seasonal dependence of $d$ (Faranda et al., 2017; Rodrigues et al., 2017). There is a clear semi-annual cycle, with peak autocorrelation
values in excess of 0.21. Over a full year there are therefore two positive and two negative peaks in autocorrelation, with the second positive peak typically displaying a larger magnitude than the first. The presence of a semi-annual cycle leads us to interpret the ACF as being modulated by the four seasons, with the first positive peak corresponding to cross-season correlation and the second, larger peak corresponding to correlation between the same seasons in successive years.

The inverse persistence, $\theta$ shows a marked variability with values ranging from 0.28 to 0.65 (i.e.1.6 to 3.6 days in terms of $1/\theta$) (Figure 1b). We note that these values should not be compared directly to the persistence of the traditional weather regimes defined using clustering algorithms, as the requirement that the flow does not leave the neighborhood of the state $\zeta$ is a more restrictive condition than continued permanence within a given cluster. Concerning the autocorrelation (Figure 1d), $\theta$ shows a very different pattern to $d$. The year-to-year correlation between same seasons is still large and positive, but the
semi-annual oscillation seen in $d$ is entirely absent. Indeed, the winter values appear to be anticorrelated to those of the other



three seasons.

An analysis of monthly mean values (Figure 2a) confirms the strong seasonal control on the dynamical characteristics of the field. In the summer months, also the variability (Figure 2b) of the two metrics reaches a minimum. With the autumn season,

both the local dimension and $\theta$ increase rapidly while the variability remains low. As winter progresses the variability increases, $\theta$ remains roughly constant while the local dimension shows a marked decrease, albeit remaining well above the summertime values. In spring $d$ grows back to values similar to those seen in autumn, while its variability peaks and $\theta$ starts decreasing. This picture is consistent with the autocorrelation functions described above. The annual cycle of $\theta$ can be explained as follows: stability peaks in summer when the mid-latitude storm tracks and wave activity are comparatively weak, decreases in autumn

and winter and starts increasing again during spring. For the annual cycle of $d$, the maxima occurring in the intermediate season can be explained as follows: assuming that there is a *winter* and a *summer* attractor, the transitional seasons are more unstable because the atmospheric flow can explore summer and winter configurations. In a more dynamical systems terminology the spring/autumn atmospheric flow seats on a sort of saddle point of the dynamics.

One can further look at the slp anomalies corresponding to extremes in $d$, here defined as events beyond the 0.02 and 0.98 percentiles of the full distribution (see dashed lines in Figure 3a). The maxima of $d$ (Figure 4a) correspond to a weakened Aleutian low and a tilted and weak Atlantic ridge-like anomaly over the North Atlantic. The minima (Figure 4b) can be explained by the dominance of hemispheric-scale modes of variability, such as the Arctic Oscillation (Faranda et al., 2016), which lead to predictable large-scale pressure patterns. We now analyze slp anomalies corresponding to extremes in $\theta$. The $\theta$ maxima

(Figure 4c) correspond to a weakened Aleutian low (as for the $d$ maxima) and a Scandinavian blocking. This configuration corresponds to large planetary-scale meanders in the mid-latitude zonal flow, i.e. to an unstable, rapidly shifting configuration. The anomalies corresponding to minima in $\theta$ (Figure 4d) closely resembles the negative phase of the Arctic Oscillation . There is a deepened Aleutian low and a negative NAO dipole - the latter having previously been associated to highly stable states (Faranda et al., 2017). The most striking feature is the presence of a strong positive anomaly at high latitudes. The $d$–$\theta$ scatter-

plot (Figure 3a) further highlights how each season forms a distinct diagonal band of highly correlated $d$ and $\theta$, suggesting that there is both inter-seasonal and intra-seasonal correlation between the two metrics. The close correspondence between the $d$ and $\theta$ extremes, marked by the dots beyond the vertical and horizontal dashed lines, explains why the large-scale slp anomaly patterns for the maxima in Figure 4a,c) resemble each other.

Figure 3(b) shows that the two metrics also have high cross-correlation values, with the lag-0 correlation approaching 0.7. The lagged cross-correlation shows a semi-yearly cycle, with the peak correlation values reflecting integer year shifts and peak anticorrelation values reflecting a shift of approximately one season. This latter feature can be easily understood in terms of the above analysis. $d$ peaks in autumn and spring, while $\theta$ has a broad peak from late autumn to early spring. If we imagine shifting the $d$ curve shown in Figure 2 forwards or backwards by one season, the broad $\theta$ peak will now match a dimension trough, thus



leading to a negative correlation. The smaller positive and negative peaks in the cross-correlation function correspond to shifts of one and three seasons, respectively, such that the two cycles are roughly in quadrature.

## 3.2 Temperature

The local dimension, $d$, of the air field shows a marked variability throughout the analysis period, with a range similar to that of

$d(\mathrm{slp})$: $8.9 < d(\mathrm{air}) < 33.3$ (Figure 5a). The average dimension $D$ is roughly 17.6, slightly lower than $D(\mathrm{slp})$. The autocorrelation function of $d$ (Figure 5c) again displays a semi-annual cycle, albeit with larger autocorrelation values than those seen for the slp. We note that these autocorrelations should not be linked directly to the large seasonal cycle in temperature, since here we are considering $d(\mathrm{air})$ which is not necessarily linked with the absolute value of the field. The inverse persistence, $\theta$ spans a range corresponding to periods between 1.9 and 6.3 days ( $0.16 < \theta < 0.54$) Figure 5b), indicating a higher persistence than

slp. $\theta$'s autocorrelation function (Figure 5d) is again different to that seen for $d$. The inter-year same-season correlation is still large and positive, but the semi-annual oscillation seen in $d$ is entirely absent. Indeed, the winter values appear to be anticorrelated to those of the other three seasons, albeit with some weak modulation in the negative correlation values on seasonal scales.

An analysis of monthly mean values (Figure 2c,d) confirms the strong seasonal control on the dynamical characteristics of

the field, but also highlights a radically different picture from that seen for the slp. In the summer months, $d$ and its variability peak while $\theta$ and its variability display a local maximum. With the autumn season, both the local dimension and $\theta$ reach a local minimum, only to increase again during wintertime. During spring, both metrics display a second minimum before returning to their high summertime values. The seasonal cycle in the variability of both indicators roughly matches that of the indicators themselves. This picture is consistent with the autocorrelation functions described above. The cycle in $d$ mirrors closely that

of $\theta$. The summertime and wintertime local maxima can be associated to the inherent difficulties in predicting the onset and end – and hence duration – of warm and cold spells (Sillmann et al., 2013; Matsueda, 2011). This is presumably linked to a high-dimensional situation with a large number of allowed preceding and future evolutions. The annual cycle of $\theta$ suggests that the winter (and to a lesser degree the summer) temperature fields are comparatively unstable, while the transitional seasons have a more sluggish dynamical evolution. This can be linked to the presence of wintertime cold spells and summertime heat

waves which are usually non-stationary and locally short-lived, although notable exceptions can occur.

One can further look at the air anomalies corresponding to extremes in $d, \theta$, again defined as events beyond the 0.02 and 0.98 percentiles of the full distribution (see dashed lines in Figure 6a). Figure 7 show the composite anomalies with respect to the seasonal cycle. The spatial anomalies corresponding to $d$ maxima (Figure 7a) are weak, suggesting that there is no single

large-scale pattern matching these extremes. The anomalies corresponding to the minima in $d$ (Figure 7b) instead show negative anomalies centred on the Arctic region, and weaker positive anomalies over Eurasia and North America. This configuration cannot be easily linked to the canonical climate variability indices, and it is not immediately obvious as to why it should be highly predictable. The $\theta$ maxima (Figure 7c) correspond to a cold Arctic and negative anomalies across the hemisphere, with a region of positive anomalies over Northern Russia and the Norwegian and Barents seas. The anomalies corresponding to





minima in $\theta$, on the contrary, (Figure 7d) display a reduced meridional temperature gradient with a warm Arctic and cold mid and low-latitudes . The winter months form a cluster corresponding to high $\theta$, relatively high $d$ values, while the spring and autumn seasons form a low $\theta$, low $d$ cluster. Summertime forms a continuation of the spring/autumn band, extending it toward higher $d$ and $\theta$ values. This marked seasonal separation leads to a weaker correspondence between the $d$ and $\theta$ extremes, and

indeed the spatial anomaly patterns shown in Figure 7 do not resemble each other between the two metrics.

The two metrics also show high cross-correlation values (Figure 6b), with the lag-0 correlation exceeding 0.5 as it is also evident from the $d$-$\theta$ scatterplot (Figure 6a), which further highlights a much more distinct seasonal separation than that seen for the slp (Figure 3a). The lagged cross-correlation shows a semi-yearly cycle, with the peak correlation values reflecting full and half-year shifts, in agreement with the synchronous double peak in both metrics shown in Figure 7. Similarly, the two large

negative peaks in the cross-correlation function correspond to shifts of one and three seasons, respectively, leading to situations in which the two yearly cycles are in anti-phase.

### 3.3 Precipitation

The local dimension, $d$, of the precipitation field (prp) shows a large variability throughout the analysis period, with markedly higher values than those of the previous variables: $26.8 < d(\text{prp}) < 83.9$ (Figure 8a). This is also reflected in the average di-

mension $D = 49.3$. These high values are consistent with the very scattered, noisy nature of the precipitation field. Again in contrast with the slp and air fields, the autocorrelation function of $d$ displays an annual cycle (Figure 8c).

The inverse persistence $\theta$ (Figure 8b) spans a range corresponding to periods between 1.0 and 2.1 days ($0.47 < \theta < 0.98$), indicating a lower persistence than the previous variables (Figure 8b). This is again linked to the very local nature of rainfall

and the very rapid variations in rainfall rate relative to those associated with, for example, pressure. $\theta$'s autocorrelation function is also different to that seen for the previous variables (Figure 8d), and indeed displays the semi-annual cycle previously seen in the autocorrelation functions of $d$. The interannual same-season correlation is still large and positive, but this is now preceded by three alternating negative and positive peaks.

An analysis of monthly mean values (Figure 2e,f) reveals a marked seasonal cycle in $d$, whose absolute values and variability both peak during the summer and into autumn, while $\theta$ displays a minimum over the same period. These are the months when the NH monsoon systems provide added large-scale complexity but comparatively persistent spatial precipitation patterns. The high persistence might also be favored by the predominantly dry summers in the Mediterranean and other mid-latitude regions, with long dry spells at regional scale being the norm. This picture is consistent with the autocorrelation functions described

above. In particular, the troughs intervening between the same-season peaks in the autocorrelation function of $\theta$ result from the summer minimum being overlaid onto the higher values seen throughout the late autumn and into spring. The weaker positive region corresponding to a 6-month shift is associated with the local $\theta$ minimum in January/February.





Due to the local and noisy nature of precipitation few features emerge from the geographical composites corresponding to extremes in the two dynamical systems metrics again defined as events beyond the 0.02 and 0.98 percentiles of the full distribution (see dashed lines in Figure 9a). $d$ maxima (Figure 10a) show a number of scattered regions with decreased precipitation especially over the East-Asian monsoon region. The largest positive anomaly is located over the eastern-central Sahel and

could be linked to anomalies in the East African rainy season, whose late Autumn peak coincides with the bulk of the high $d$ extremes. $d$ minima (Figure 10b) show a zonally elongated dipole anomaly across the Pacific, with drier conditions in the tropics and enhanced precipitation roughly coincident with the storm track latitudes. $\theta$ maxima (Figure 10c), on the contrary, display predominant dry anomalies over the Pacific basin but relatively weak changes over continental land masses. Finally, minima in $\theta$ (Figure 10d) display a large positive precipitation anomaly over the Western Pacific, perhaps indicating a shift in

the climatological subtropical high located in the region, and weakened precipitation across northern China and in the western tropical Pacific.

The $d$-$\theta$ scatterplot (Figure 9a) shows a broad cloud of points and a clear separation between the summer, autumn and winter/spring seasons . The summer months form a cluster corresponding to relatively large $d$ but low $\theta$. The autumn cluster

also displays large $d$ values, now associated with high $\theta$s. Finally, the winter/spring cluster comprises the near-totality of low $d$ values. The two prp metrics show weaker cross-correlation (Figure 9b) values than those previously discussed, with the lag-0 correlation barely exceeding 0.1. The lagged cross-correlation shows a yearly cycle, with the peak correlation values reflecting a roughly 5-month shift and the peak anti-correlation values corresponding to a roughly 10-month shift. This results from the asymmetry in the duration of the peaks and troughs in the two variables, which preclude any seasonal variation in the

cross-correlation from becoming evident.

## 4   Cross-analysis of the dynamical properties

We next address the co-variability of the dynamical indicators of the different variables. In physical space, there is an obvious link between anomalies in the large-scale slp and near-surface temperature fields. A similarly close link can be found between precipitation and temperature or slp anomalies. There are therefore strong grounds to expect some systematic relationships to

emerge.

We begin by analysing the cross-correlation functions between the slp and air (Figure 11a,b). $d$(slp) and $d$(air) are anti-correlated at zero lag, as might be expected by their contrasting seasonal cycles described above. The lagged cross-correlations display a roughly regular semi-yearly cycle, which derives from the fact that both local dimensions display a double peak, albeit

in different seasons. The cross-correlation between $\theta$(slp) and $\theta$(air) is more nuanced, owing to the fact that $\theta$(slp) displays a single yearly peak while $\theta$(air) displays two, one of which partially overlaps with the high autumn and winter values seen in $\theta$(slp). The lag-0 correlations are positive, albeit low, and peak negative cross-correlations are achieved at lags of approximately 6-7 months. The lag-0 anticorrelation of the local dimensions points to the fact that it is rare to find co-occurring slp





and air fields both displaying high predictability. This is compounded by the fact that positive correlations between the $\theta$ are generally weak, suggesting that persistent slp configurations do not necessarily match equally persistent air patterns. An example of this are wintertime cold spells in the mid-latitudes: while the large-scale circulation anomalies are often very persistent, the temperature evolution is highly non-stationary with a build-up of cold air masses leading to a rapidly cooling region which

then relaxes back to near-climatological values as soon as the anomalous circulation pattern weakens (Messori et al., 2016).

The cross-correlation between $d(\text{slp})$ and $d(\text{prp})$ is shown in Figure 11(c,d). At lag-0, the two variables have a weak positive correlation, with peak positive values being reached for shifts of 1-2 months. Indeed, $d(\text{prp})$ has a broad peak during the monsoon season and then decreases rapidly through the winter season, such that its late winter minimum almost matches the

early winter minimum in $d(\text{slp})$. However, we note that peak cross correlation values are significantly lower than those seen bewteen $d(\text{slp})$ and $d(\text{air})$. The two persistence metrics, on the contrary, are highly correlated at lag-0, since they both display a broad region of high values between fall and spring and a minimum during summer. The large-scale circulation changes associated with the onset of the monsoonal precipitation over Asia and Africa therefore do not seem to influence $d(\text{slp})$ in a major fashion, while the high summertime persistence in the precipitation field – presumably linked to the monsoon systems

and the predominantly dry summers over other mid-latitude regions – is matched by a high persistence in the slp field. This disconnect between persistence and predictability derives predominantly from the precipitation field, since the two metrics are highly correlated at 0 lag for the slp.

The air–prp pair is analysed in Figure 11(e,f). In this case, the lag-0 cross-correlation between the local dimensions is almost

exactly 0, since the autumn trough and late winter peak in $d(\text{air})$ match the autumn peak and late winter trough in $d(\text{prp})$. A similar picture is seen for the persistence metrics, since the summer peak in $\theta(\text{air})$ is almost perfectly out of phase with the summer minimum in $\theta(\text{prp})$. The large temperature changes associated with the onset of the monsoonal precipitation over Asia and Africa therefore do not seem to directly influence $d(\text{slp})$ nor the persistence of the large-scale temperature fields.

The cross-correlations consider the timeseries of the different metrics as a whole, but provide little insights on the correlation between dynamical extremes. We conclude our analysis by looking at the $d$-$\theta$ scatterplots for the local dimensions and persistences of the three observables (Figure 12). The negative lag-0 correlation found for $d(\text{slp})$ and $d(\text{air})$ is evident (Figure 12a), while the other two $d$ scatterplots (Figure 12c,e) show a more diffuse distribution, consistent with the low correlation values previously discussed. However, in all three cases there is no clear correspondence between the dynamical extremes. $\theta$

shows a similar picture for the air–prp case (Figure 12f), While the slp–air (Figure 12b) and slp–prp (Figure 12d) pairs show some co-occurrence of high-$\theta$ (and hence low persistence) episodes. This indicates that rapidly shifting slp patterns can lead to equally rapid shifts in the large-scale temperature and precipitation fields.



## 5 Conclusions

In the present study we have applied recent advances in dynamical systems theory to estimate local dimension and inverse persistence of instantaneous atmospheric fields over the Northern Hemisphere. Persistence is a very intuitive metric, which quantifies the average residence time of the system's trajectory in phase space within the neighbourhood of the point of inter-
est. Local dimension is a proxy for the number of locally active degrees of freedom in the system, and can thus be directly linked to the number of possible configurations preceding and following the instantaneous field being analysed. We have specifically focused on three observables: sea-level pressure, 1000 hPa temperature and precipitation rate. Despite the high dimensionality of atmospheric dynamics, we find that the Northern Hemisphere sea level pressure and low-level temperature fields can on average be described by roughly twenty degrees of freedom, while the noisier precipitation field has an average dimension of
almost 50. We further note that the dimension of the instantaneous fields can vary by almost a factor of 4 for a given observable. The links between the local dimension and persistence of a given variable can be complex. While the two generally show a positive correlation – albeit very weak for precipitation – they can display relatively different seasonal cycles.

This study further analyses dynamical extremes, namely the instances where one – or both – dynamical systems metrics is
at the positive or negative edge of its distribution. The dynamical extremes in $d$ and $\theta$ of a given variable often occur independently. For example, maxima in $d$ of 1000 hPa temperature almost never coincide with maxima in $\theta$ for the same variable. Both $d$ and $\theta$ are linked to atmospheric predictability, since a persistent, low-dimensional state is intrinsically easier to forecast than a rapidly shifting, high-dimensional situation. Fields where the co-occurrence of $d$ and $\theta$ extremes is more frequent – such as is the case for slp – therefore provide more highly predictable (or unpredictable) configurations than those where the two
occurrences are mutually exclusive.

We further identify a number of robust correlations between the dynamical properties of the different variables. For example, low-persistence cases in slp often indicate a similarly low persistence in air and prp. This is an intuitive relationship since rapidly shifting slp patterns can lead to equally rapid shifts in the large-scale temperature and precipitation fields. Other links
do not have a similarly straightforward physical interpretation. For example, the local dimensions of precipitation and slp seem to be mostly uncorrelated, suggesting that the large monsoonal precipitation phenomena seen across the NH do not directly affect the predictability of the large-scale slp field.

We conclude that the dynamical systems metrics we adopt here provide a wealth of information concerning the large-scale
atmospheric processes and dynamics. We are convinced that this analysis framework will find applications in a wide number of climate studies.

*Code availability.* The code to perform the analysis is available upon request to the corresponding author.





*Data availability.* The dataset NCEP-NCAR is public available at https://www.esrl.noaa.gov/psd/data/gridded/data.ncep.reanalysis.html

*Author contributions.* DF and GM performed the analysis. MCAC downloaded and organised the datastes. All the authors contributed to the writing.

*Competing interests.* No competing interests are present

5   *Acknowledgements.* P.Yiou, D. Faranda and G Messori were supported by ERC grant No. 338965, M.C. Alvarez-Castro was supported by Swedish Research Council grant No. C0629701 and G. Messori was supported by a grant from the Department of Meteorology of Stockholm University.



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



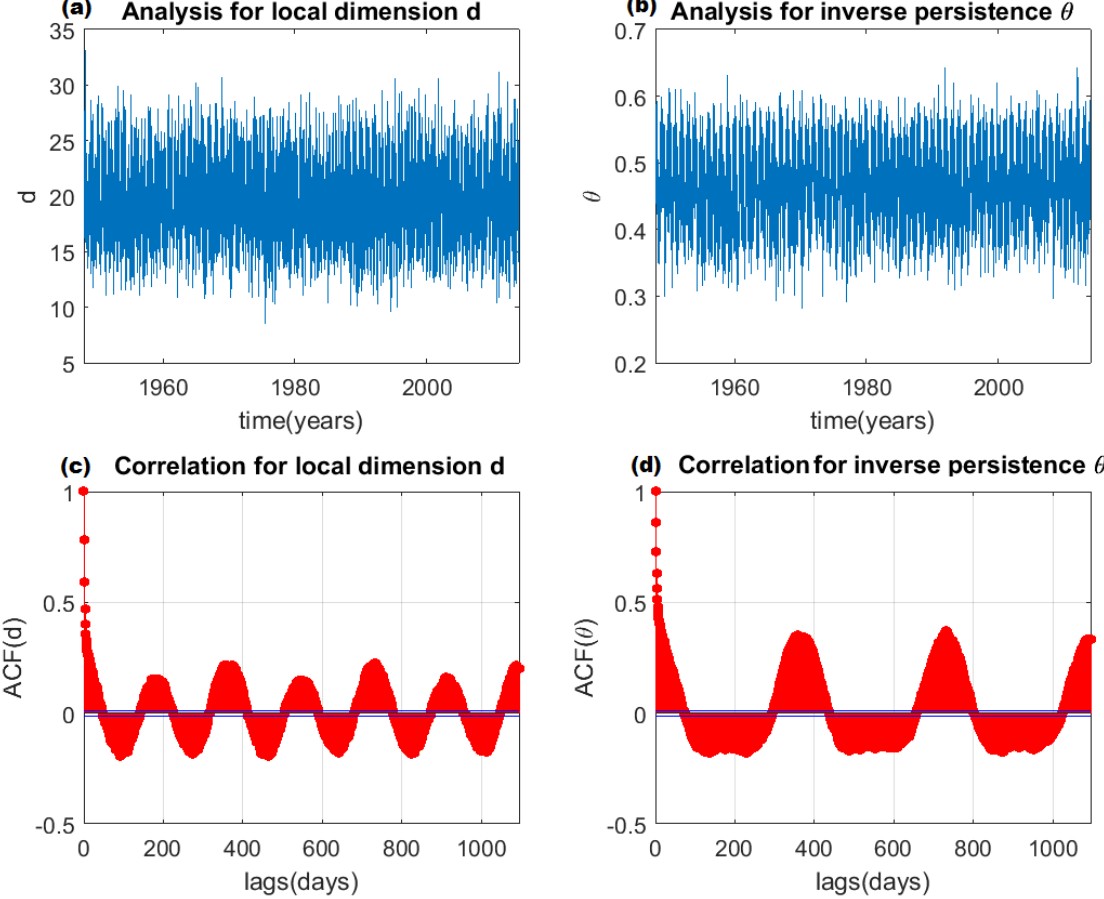

**Figure 1.** Statistics of local dimension $d$ and persistence $\theta$ for daily sea-level pressure (slp) data from the NCEP-NCAR reanalysis. Time series of daily values of $d$ (a) and $\theta$ (b). autocorrelation function ACF($d$) (c) and ACF($\theta$) (d) for three years in daily lags.



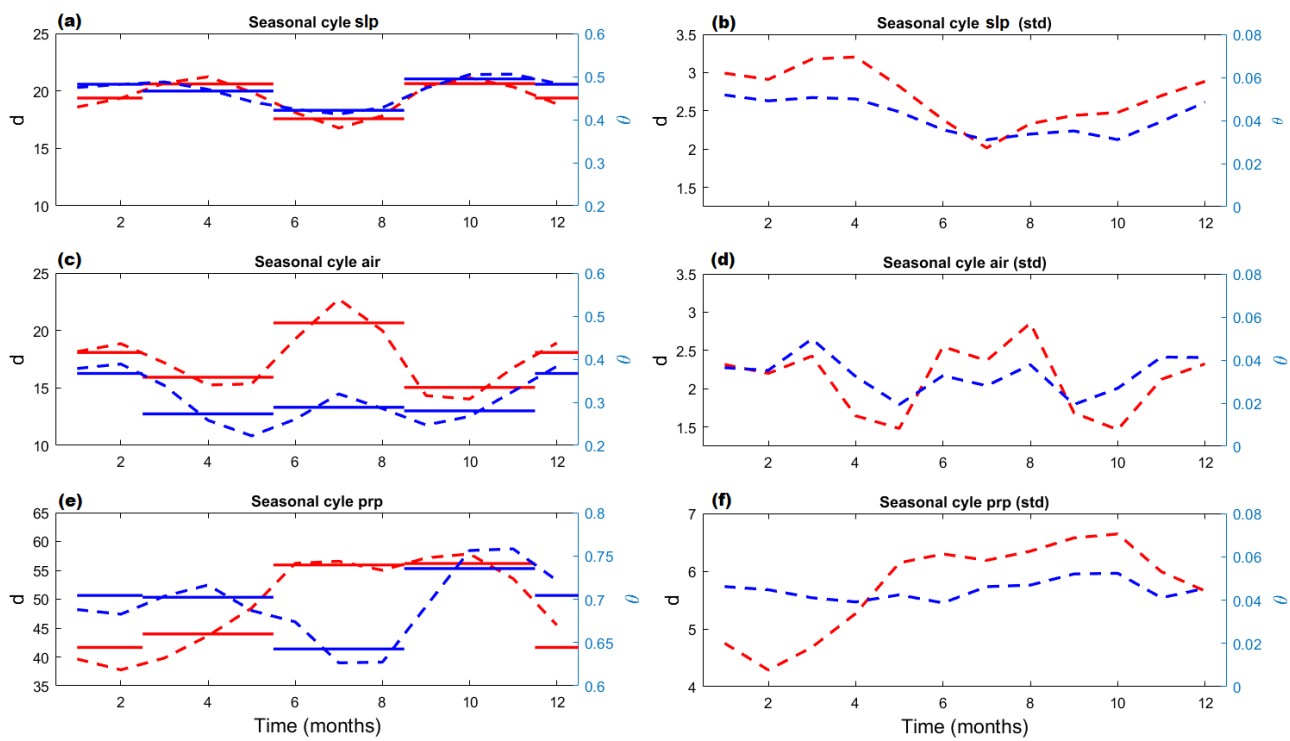

**Figure 2.** Monthly (dotted lines) and seasonal (continuous lines) average values (a,c,e) and standard deviation (b,d,f) for the local dimension $d$ (red) and inverse persistence $\theta$ (blue) of sea level pressure slp (a,b), temperature at 1000 hPa air (c,d) and precipitation rate prp (e,f) data.

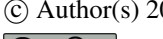


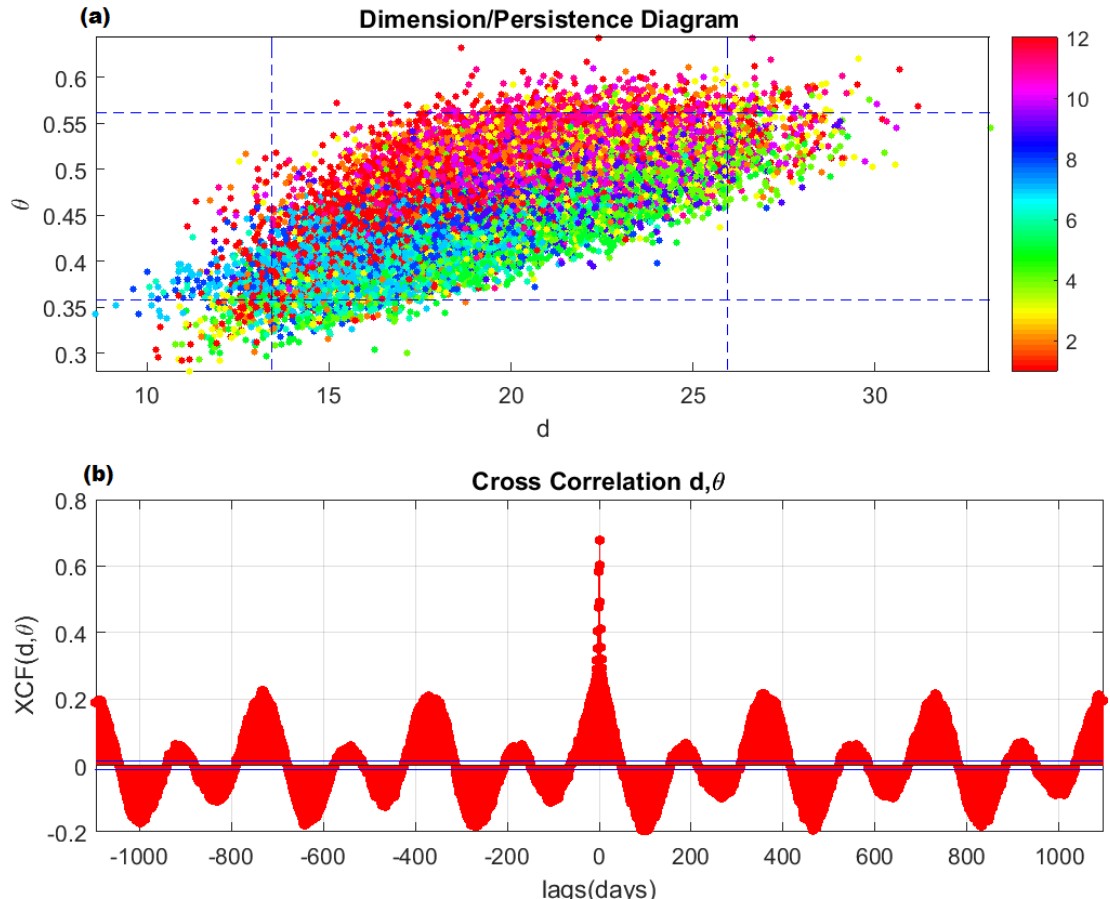

**Figure 3.** (a) Scatter-plot of local dimension $d$ and inverse persistence $\theta$ for slp data. Each points represents the value corresponding to one day in the NCEP-NCAR reanalysis. The color indicates the month of the year the datapoint fall in. Blue dotted lines indicate the 0.02 and 0.98 percentiles of the $d, \theta$ distributions. (b) Cross correlation function between Local dimension $d$ and inverse persistence $\theta$ for daily sea-level pressure (slp) data from the NCEP-NCAR reanalysis.




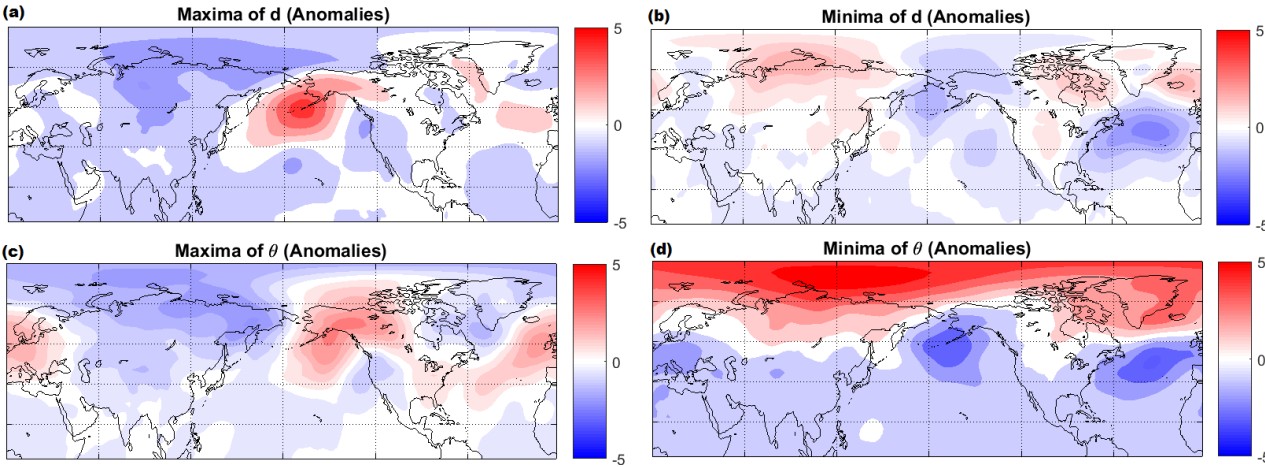

**Figure 4.** Composite anomalies with respect the seasonal cycle in sea-level pressure (slp) for the four phase-space regions delimited by the blue dotted lines in Fig 3. maxima of $d$ (a), minima of $d$ (b), maxima of $\theta$ (c), minima of $\theta$ (d). Units: hPa




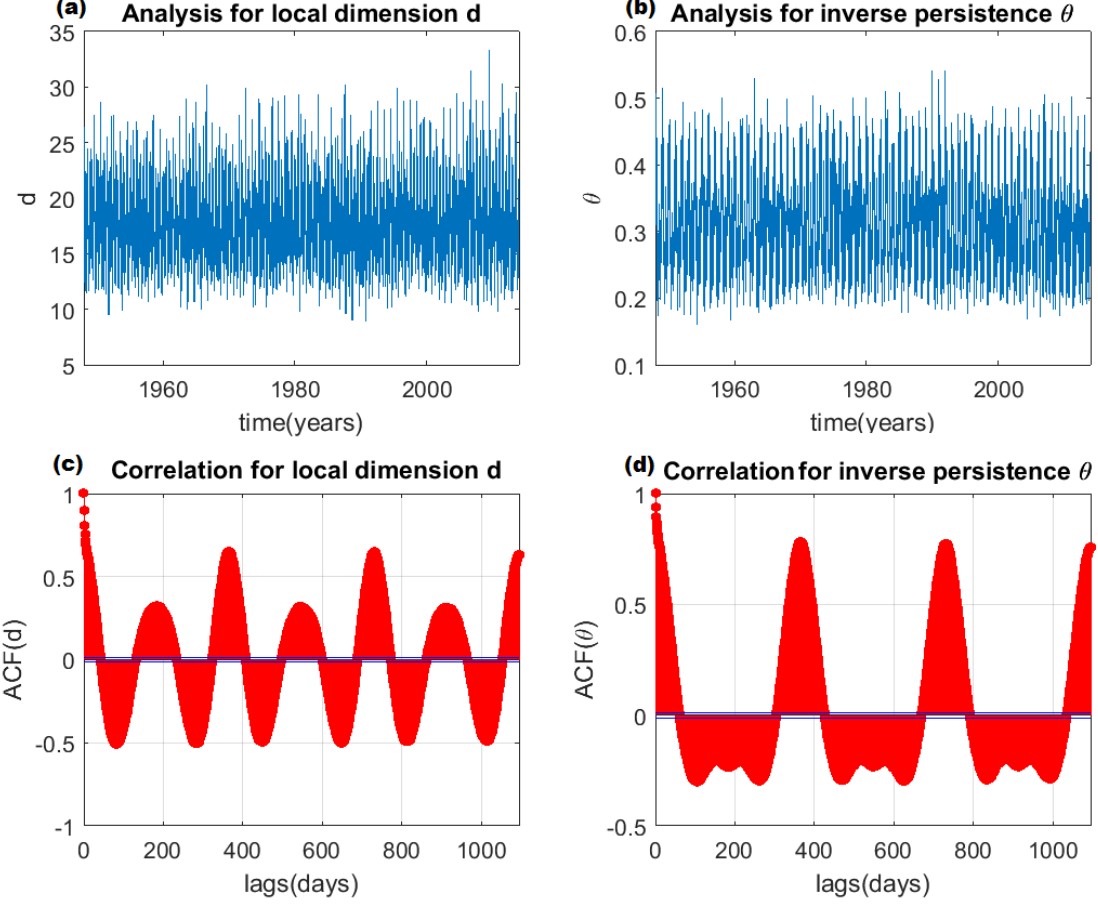

**Figure 5.** Statistics of local dimension $d$ and persistence $\theta$ for daily temperature at 1000 hPa (air) data from the NCEP-NCAR reanalysis. Time series of daily values of $d$ (a) and $\theta$ (b). Autocorrelation function ACF($d$) (c) and ACF($\theta$) (d) for three years in daily lags.

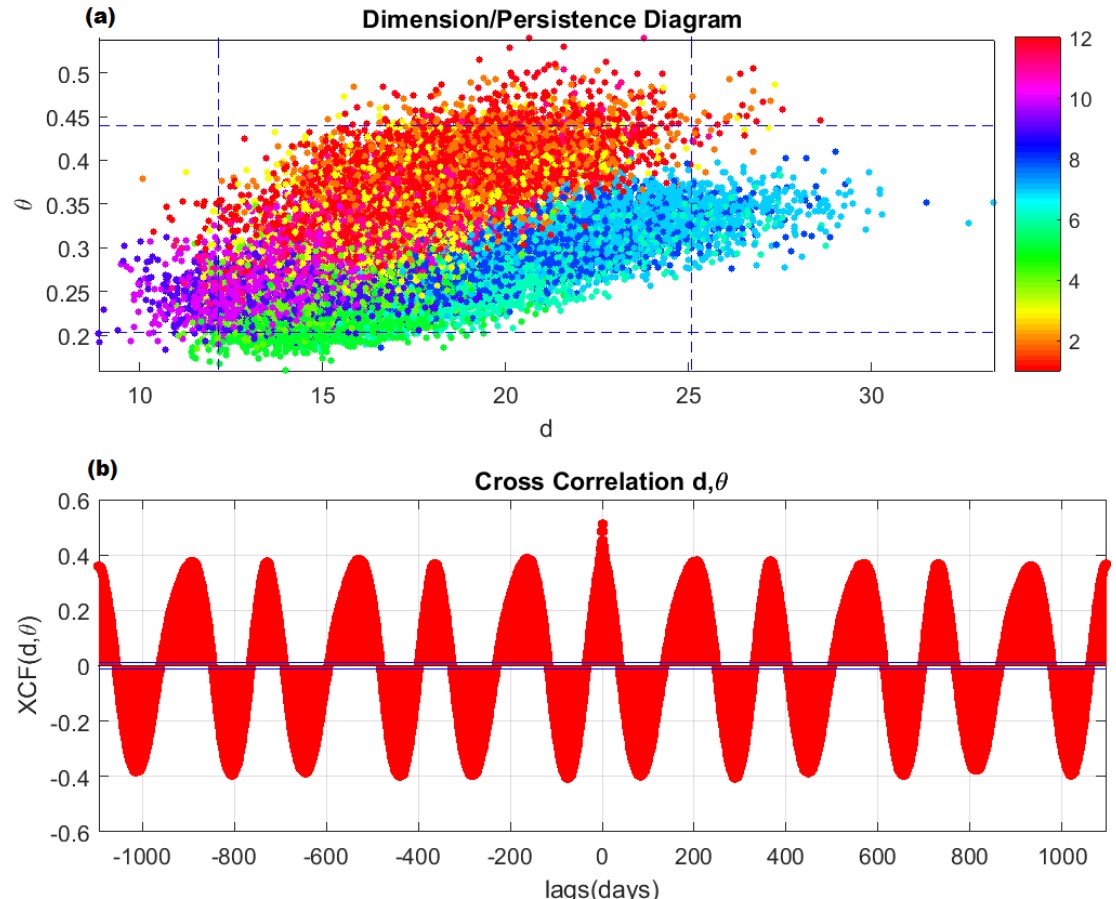

**Figure 6.** (a) Scatter-plot of local dimension $d$ and inverse persistence $\theta$ for daily temperature at 1000 hPa (air) data. Each points represents the value corresponding to one day in the NCEP-NCAR reanalysis. The color indicates the month of the year the datapoint fall in. Blue dotted lines indicate the 0.02 and 0.98 percentiles of the $d, \theta$ distributions. (b) Cross correlation function between Local dimension $d$ and inverse persistence $\theta$ for daily temperature at 1000 hPa (air) data from the NCEP-NCAR reanalysis.

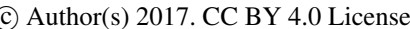



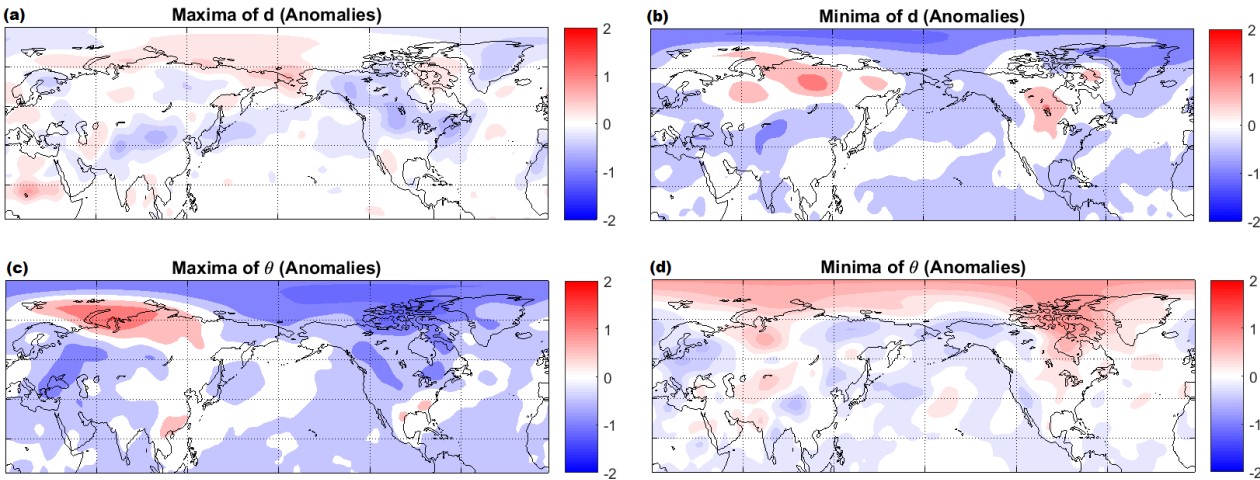

**Figure 7.** Composite anomalies with respect the seasonal cycle in daily temperature at 1000 hPa (air) for the four phase-space regions delimited by the blue dotted lines in Fig 6. Maxima of $d$ (a), minima of $d$ (b), maxima of $\theta$ (c), minima of $\theta$ (d). Units: $^{\circ}$ C





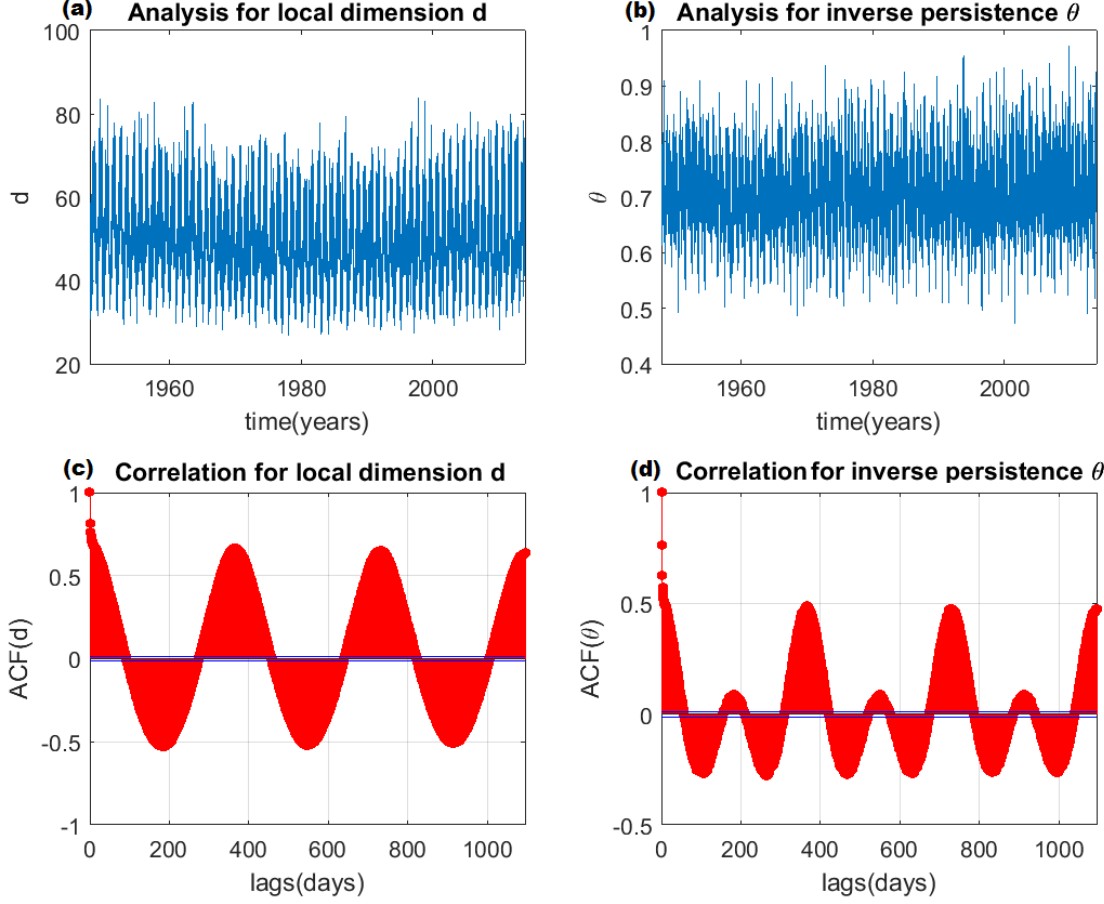

**Figure 8.** Statistics of local dimension $d$ and persistence $\theta$ for precipitation rate (prp) data from the NCEP-NCAR reanalysis. Time series of daily values of $d$ (a) and $\theta$ (b). Autocorrelation function ACF($d$) (c) and ACF($\theta$) (d) for three years in daily lags.

<

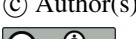


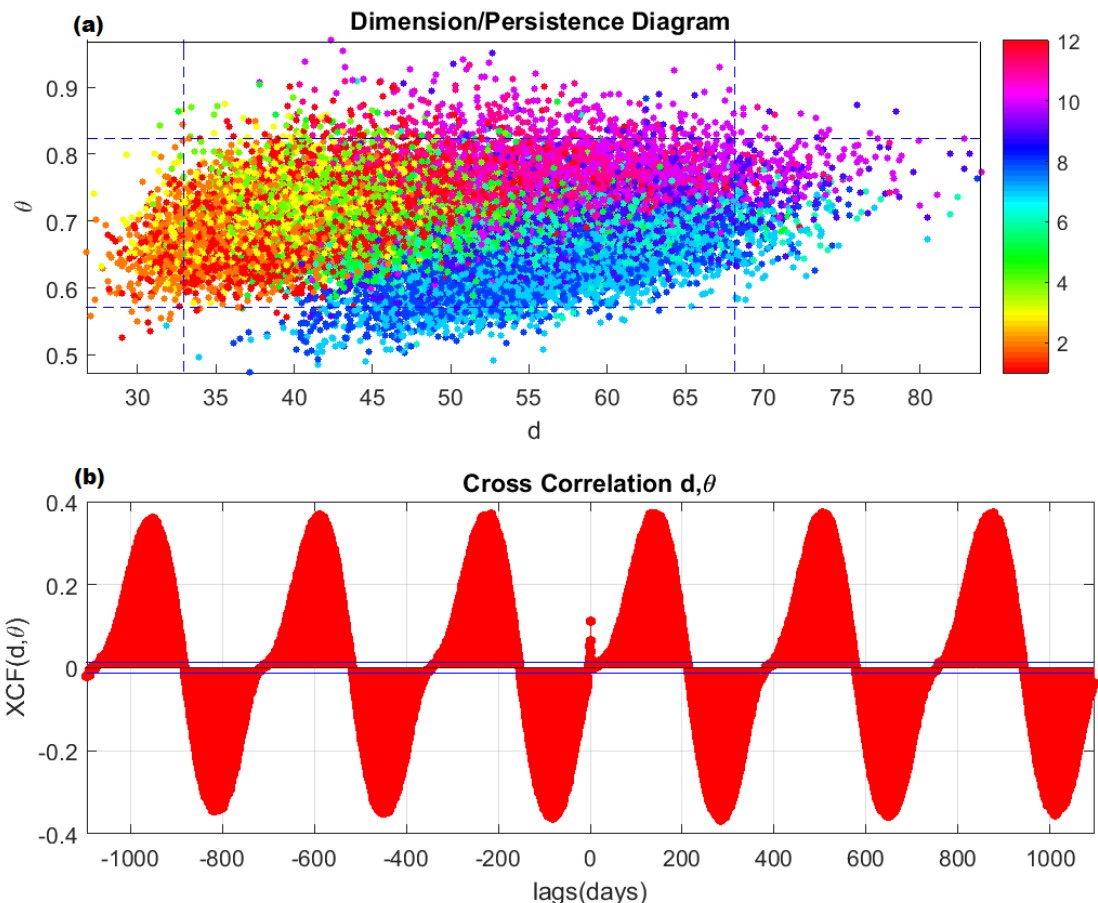

**Figure 9.** (a) Scatter-plot of local dimension $d$ and inverse persistence $\theta$ for prp data. Each points represents the value corresponding to one day in the NCEP-NCAR reanalysis. The color indicates the month of the year the datapoint fall in. Blue dotted lines indicate the 0.02 and 0.98 percentiles of the $d, \theta$ distributions. (b) Cross correlation function between Local dimension $d$ and inverse persistence $\theta$ for precipitation rate (prp) data from the NCEP-NCAR reanalysis.



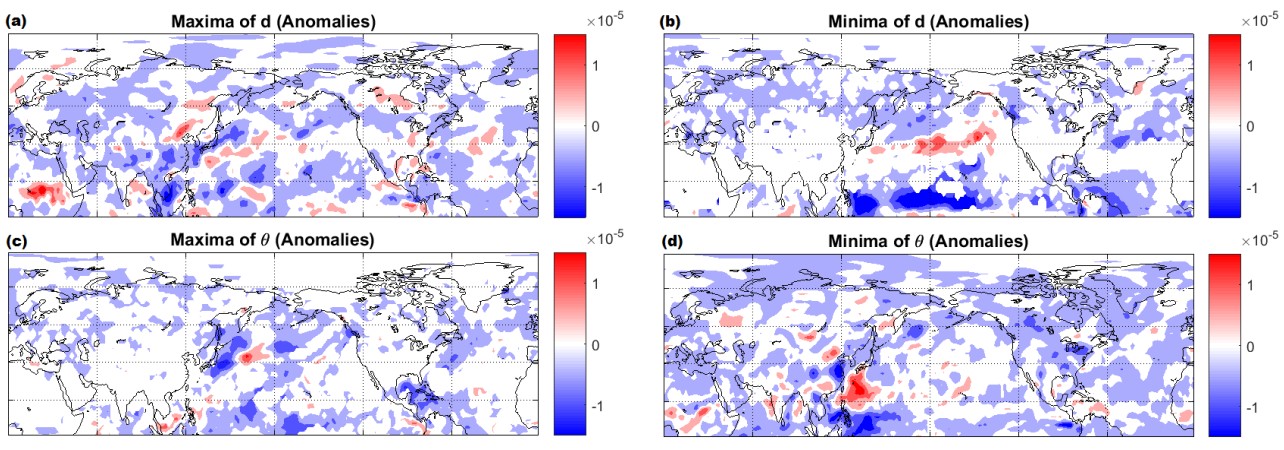

**Figure 10.** Composite anomalies with respect the seasonal cycle in precipitation rate for the four phase-space regions delimited by the blue dotted lines in Fig 9. Maxima of $d$ (a), minima of $d$ (b), maxima of $\theta$ (c), minima of $\theta$ (d). Units: Kg/m$^2$/s (mm/s)



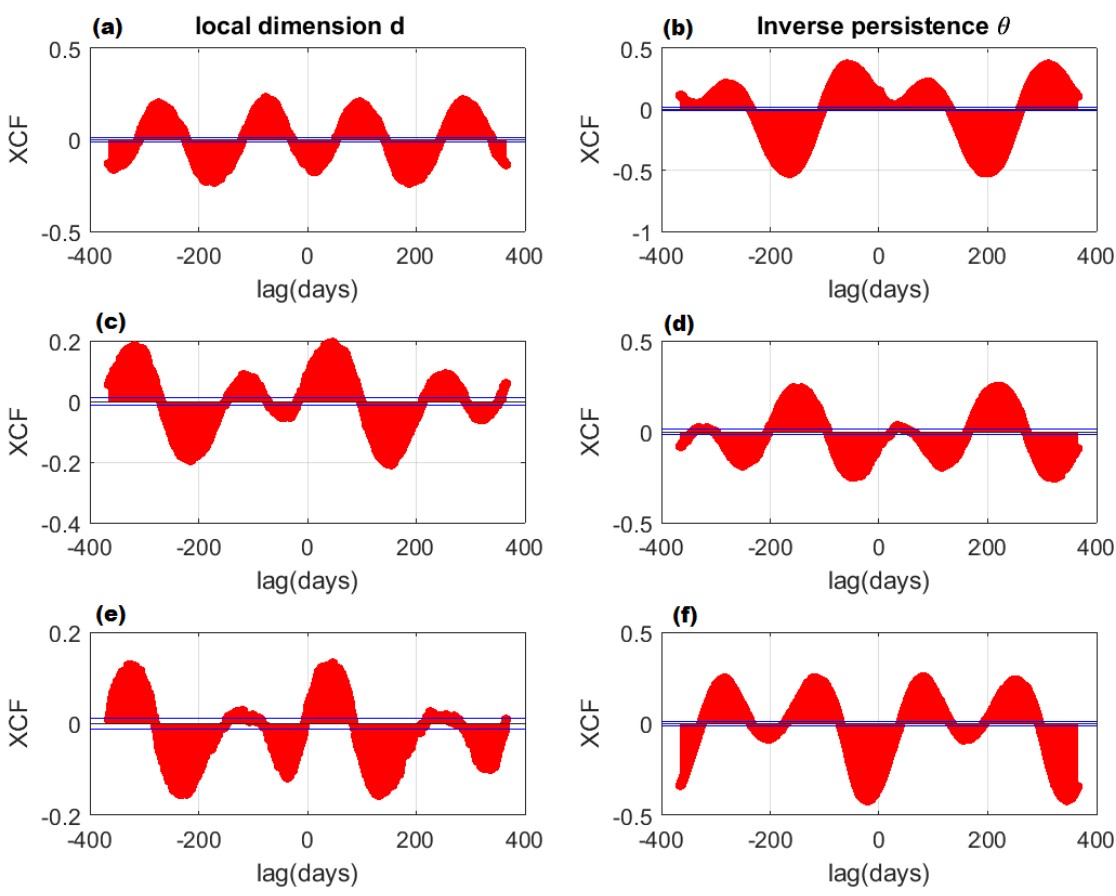

**Figure 11.** Cross correlation functions for the local dimension $d$ (a,c,e) and inverse persistence $\theta$ (b,d,f) of slp, air and prp data.



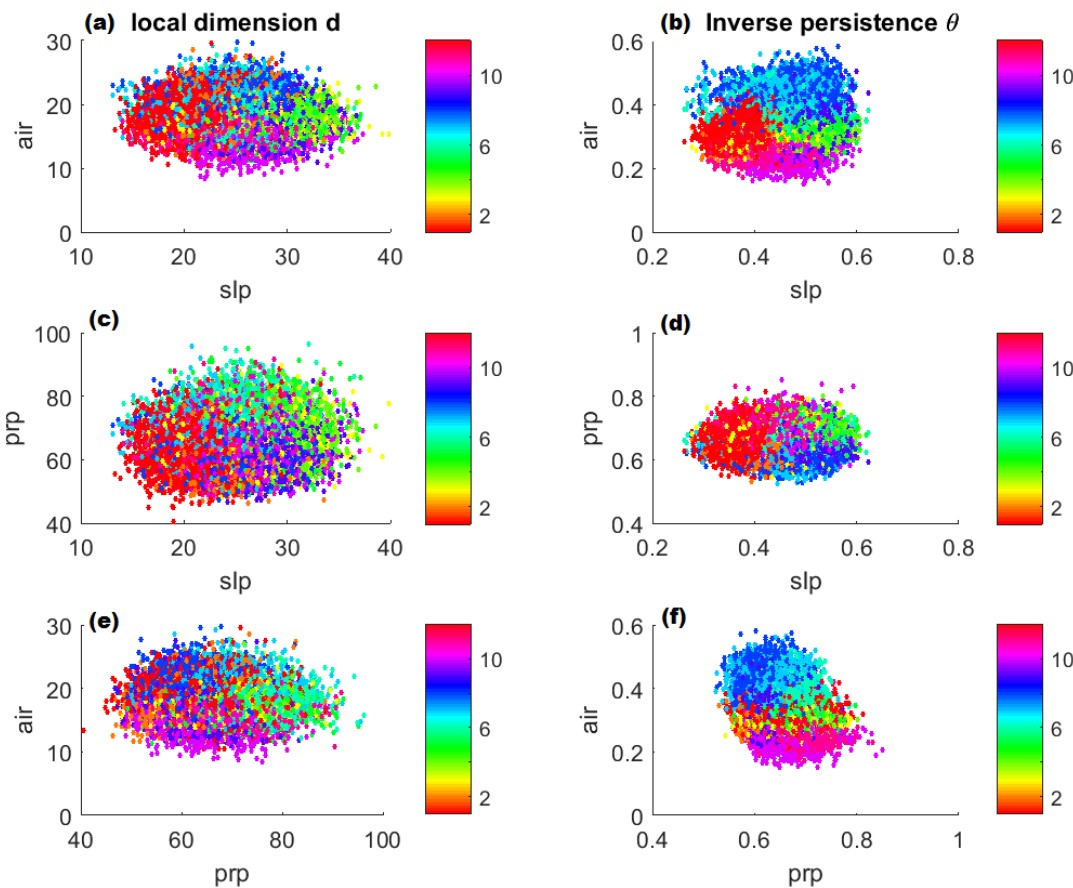

**Figure 12.** Scatter plots of the local dimensions $d$ (a,c,e) and inverse persistences $\theta$ (b,d,f) between sea level pressure (slp), temperature at 1000 hPa (air) and precipitation rate (prp) data.