# Peer review of "DYNAMICAL PROPERTIES AND EXTREMES OF NORTHERN HEMISPHERE CLIMATE FIELDS OVER THE PAST 60 YEARS"

_Nonlinear Processes in Geophysics, 2017_

## Referee Comment (RC1) · Anonymous Referee #1 · 11 Aug 2017

Authors present an important application of the of the extreme value theory to Poincaré recurrences in dynamical systems, which can provide estimations of some metrics of the ergodic chaotic attractors, namely the phase-space local dimension and local recurrence. Authors apply the technique to relevant daily fields: the sea level pressure, surface air temperature and precipitation rate, trying to understand their seasonal dependency in the Northern Hemisphere and exploring the atmospheric patterns which are consistent with maximum and minimum values of those metrics. There are, however some minor worthwhile points which should be addressed before publication.

1 - Annual cycles of the local dimension d for slp and Temp(air) are in phase opposition

[Figure]

(Figs. 2a, 2c). There is any physical reasonable explanation for that? The authors present a hypothetical explanation for the minimum of d for slp at the summer and maximum at the equinoxes as being more 'unstable' because of an incremented exploration of the summer and winter configurations. However, that could apparently be valid for any field but in fact it is totally contradicted in the case of the Temp(air) where d is higher in summer. Therefore, the authors should reformulate the explanation (which, as presented likes somehow ad-hoc) or give and explanation why it does not work for Temp(air).

2 – In Pg. 8 in the Analysis of Precipitation data. Authors explain the higher value of d (compared to slp and air) thanks to the scattered and noisier character of the precipitation field. However, analysis of precipitation extremes has been done using the multifractal analysis (MFA) (Langousis et al. 2009) which also give hints about the dimension. Authors should try to relate the presented results with those of MFA.

3 - The graphs of ACFs(s) (Fig.1 and others) present discontinuities or overlaps near the zero line. This is maybe because of the symbol size? Use a format avoiding that artifact.

4 - Caption of Fig. 11 shall be more complete, i.e. making explicit the pairs of variables: Fig11a,b (slp-air), Fig. 11c,d (slp-prp), Fig. 11e,f (air-prp). Which variable lags in the future?

LANGOUSIS, A., VENEZIANO, D., FURCOLO, P. y LEPORE, C. (2009): ÂńMultifractal rainfall extremes: Theoretical analysis and practical estimationÂż. Chaos, Solitons and Fractals, n° 39, 1182-1194.

---

## Referee Comment (RC2) · Anonymous Referee #2 · 10 Sep 2017

The authors analyze dynamical properties of daily surface pressure, surface temperature and precipitation rate using two parameters from a universal distribution derived in previous research from extreme value theory of Poincaré recurrences. The two parameters, local dimension of the phase space attractor and the inverse of the average residence time, are physically intuitive, and have the potential to characterize predictability of the analyzed variables.

The manuscript is of interest for both the Dynamical System and Applied Climatology scientific communities, illustrating an approach that could be used in other regions of the world and for other variables of interest. Nonetheless, the analysis raises three

main issues that I think need to be clarified before the paper can be considered for publication in NLPG. In addition, a number of minor changes/clarifications are also suggested to the authors.

**Main issues**

The manuscript identifies seasonality as a key regulator of the behavior of local dimension and persistence. For example, for surface pressure, two maxima and minima per year are reported, in seasons considerably different (e.g., the typical associated teleconnections are not the same in the two max seasons, with potential implications for the phase-space trajectories and their characteristics). Nonetheless, my understanding is that the composite of anomalies shown in Figure 4 for the maxima and minima are computed from the "extreme" regions in the dimension/persistence diagram (Figure 3, in the example considered here), without discriminating between the two seasons. As a result, I wonder if the present composite analysis is not averaging fields that correspond to different situations. Please clarify this point and the possible differences that could be obtained if analyzing separately the two different maxima and minima.

My second major concern is somehow related to the first one. Although something similar might be happening for precipitation rate (there is only one maximum, but it is so long that different climate agents might be involved, and thus perhaps it makes sense to try to sub-sample that season), I wonder if less noisier results could be obtained if a different precipitation variable is used. As the authors are probably aware, precipitation rate is in general too noisy and far less predictable than accumulated rainfall or rainfall frequency, the latter being the most predictable of all three –at least in most parts of the world. How much Figure 10 changes if frequency of precipitation is used?

I also think that it is adequate that the authors include at least one comment on the fact that considering all grid boxes in the Northern Hemisphere might be "masking" the predictability and/or the dynamical properties of the variables in study, compared to performing the analysis in regions that are known to have a more homogeneous

climatic response to different sources of predictability. In other words: how different are the results if instead of considering the entire Northern Hemisphere, regions of homogenous predictability are considered? Seasonality is different for different regions, so is it possible that mixing the seasonal cycle of so many places degrades the characterization of the dynamical properties of the system?

**Minor details** (P indicates page number, and L the line number)

P2L12: excise "the" before "attractor"

P2L33: how many are "enough"?

P2L34-P3L1: something is wrong. There is a "." misplaced, and I think the trajectory is around $\xi$, not around $\theta(\xi)$, which has not been defined yet.

P3L7: I suggest to change "air" by something else. Several of the readers of the paper are going to be fast-readers, and in several other places "air" might be misleading. Perhaps use "t2m", which is one of the standards.

P3L8: what do the authors mean by "peculiar subspace"? Clarify.

P5L2: further explain what is the "length" of a cluster.

P5L15-16: excise the subjective phrasing "a very".

P5L22: maybe the semi-annual cycle is inherited from some semi-annual variability phenomena too? e.g., North Atlantic Sub-tropical High?

P5L28: why is it more restrictive?

P6L16-17: it seems to me that the weakened Aleutian low is actually a high in that figure. Can you comment on that? Also, the magnitudes of the anomalies are really low (amplitude of 10 hPa). This might be associated, again, with the fact that all days are considered together, without seasonal discrimination. The Aleutian low is typical of winter; can the authors comment on why it should be expected to appear during the

maxima of d?

P6L26: please explain why that means intra-seasonal correlation.

P7L11: any hypothesis for why is it absent?

P7L22: explain what do you mean by "high-dimensional situation".

P7L30: it is a bit strange that extremes cannot be matched to large-scale patterns, as they tend to be controlled by synoptic configurations. Maybe what the authors means is that there is no global pattern associated with the extremes? I do not expect that case either. Right now, the sentence is confusing.

P8L12: "Precipitation rate"

P11L9-10: use "twenty" and "fifty", or "20" and "50".

P11L16: along the paper, the authors use 1000 hPa. Did they interpolated? or they just mean "surface". If the latter, please change to "surface".

P11L26-27: I actually expect to see a correlation between pressure gradients and monsoonal precipitation. Maybe I'm not understanding something here.

---

## Author Comment (AC1) · 31 Oct 2017

QUERY: Authors present an important application of the of the extreme value theory to Poincaré recurrences in dynamical systems, which can provide estimations of some metrics of the ergodic chaotic attractors, namely the phase-space local dimension and local recurrence. Authors apply the technique to relevant daily fields: the sea level pressure, surface air temperature and precipitation rate, trying to understand their seasonal dependency in the Northern Hemisphere and exploring the atmospheric patterns which are consistent with maximum and minimum values of those metrics. There are, however some minor worthwhile points which should be addressed before publication.

[Figure]

ANSWER: We thank the Referee for the positive comments. We provide a point-by-point reply below.

QUERY: 1 – Annual cycles of the local dimension d for slp and Temp(air) are in phase opposition (Figs. 2a, 2c). There is any physical reasonable explanation for that? The authors present a hypothetical explanation for the minimum of d for slp at the summer and maximum at the equinoxes as being more 'unstable' because of an incremented exploration of the summer and winter configurations. However, that could apparently be valid for any field but in fact it is totally contradicted in the case of the Temp(air) where d is higher in summer. Therefore, the authors should reformulate the explanation (which, as presented looks somehow ad-hoc) or give and explanation why it does not work for Temp(air).

ANSWER: This is an interesting question that motivated some further discussion. Eventually, we concluded that the t2m dynamics fits into a single potential well dynamics, with the extremes located in winter and in summer. We can imagine a sort of Langevin model for slp-t2m, where the slp is the variable pushed in the "winter" or "summer" potential wells, while the temperature acts as forcing noise term (i.e. single potential well dynamics) with extremes in summer and winter. We have now added this possible explanation to Sect. 3.2 of the manuscript.

QUERY: 2 – On pg. 8 in the Analysis of Precipitation data. Authors explain the higher value of d (compared to slp and air) thanks to the scattered and noisier character of the precipitation field. However, analysis of precipitation extremes has been done using the multifractal analysis (MFA) (Langousis et al. 2009) which also give hints about the dimension. Authors should try to relate the presented results with those of MFA.

ANSWER: We have read with interest the paper suggested by the Referee and included it in the discussion of prp results. We have also replaced the precipitation rate with the precipitation frequency, as suggested by Referee #2. This has led to a radical restructuring of the section analyzing the precipitation data.

QUERY: 3 – The graphs of ACFs(s) (Fig.1 and others) present discontinuities or over-laps near the zero line. This is maybe because of the symbol size? Use a format avoiding that artifact.

ANSWER: Thanks for spotting this, the original ACF figures have been replaced with ones displaying the values with cross-shaped markers to avoid artefacts.

QUERY: 4 – Caption of Fig. 11 should be more complete, i.e. making explicit the pairs of variables: Fig11a,b (slp-air), Fig. 11c,d (slp-prp), Fig. 11e,f (air-prp). Which variable lags in the future?

ANSWER: We have updated the panels in that figure that currently is figure 13, and we have explained which variables lag in the future.

---

## Author Comment (AC2) · 31 Oct 2017

QUERY: The authors analyze dynamical properties of daily surface pressure, surface temperature and precipitation rate using two parameters from a universal distribution derived in previous research from extreme value theory of Poincaré recurrences. The two parameters, local dimension of the phase space attractor and the inverse of the average residence time, are physically intuitive, and have the potential to character­ize predictability of the analyzed variables. The manuscript is of interest for both the Dynamical System and Applied Climatology scientific communities, illustrating an ap­proach that could be used in other regions of the world and for other variables of interest. Nonetheless, the analysis raises three main issues that I think need to be clarified before the paper can be considered for publication in NLPG. In addition, a number of minor changes/clarifications are also suggested to the authors.

ANSWER: We thank the referee for the detailed comments, which we have addressed as detailed below.

QUERY: The manuscript identifies seasonality as a key regulator of the behavior of local dimension and persistence. For example, for surface pressure, two maxima and minima per year are reported, in seasons considerably different (e.g., the typical associated teleconnections are not the same in the two max seasons, with potential implications for the phase-space trajectories and their characteristics). Nonetheless, my understanding is that the composite of anomalies shown in Figure 4 for the maxima and minima are computed from the "extreme" regions in the dimension/persistence diagram (Figure 3, in the example considered here), without discriminating between the two seasons. As a result, I wonder if the present composite analysis is not averaging fields that correspond to different situations. Please clarify this point and the possible differences that could be obtained if analyzing separately the two different maxima and minima.

ANSWER: The referee is right about the seasonal dependence of the results. We have now added a new Figure 3 showing the distribution of dynamical extremes for every season. We note that in most cases the bulk of the maxima or the minima of the dimension/persistence corresponds to a single season. This is however not true for SLP, which shows a greater spread between seasons. For all the cases where two or more seasons have > 25% of the extremes, we have therefore produced separate composites for each season. These are shown in the new Fig 3. We have further added a sign test to all geographical composites to display the regions where more than 2/3 of the composite members agree on the sign of the anomalies. We have extensively modified the manuscript in Section 3 to account for the updated analysis and the new figures.

QUERY: My second major concern is somehow related to the first one. Although something similar might be happening for precipitation rate (there is only one maximum, but it is so long that different climate agents might be involved, and thus perhaps it makes sense to try to sub-sample that season), I wonder if less noisier results could be obtained if a different precipitation variable is used. As the authors are probably aware, precipitation rate is in general too noisy and far less predictable than accumulated rainfall or rainfall frequency, the latter being the most predictable of all three – at least in most parts of the world. How much would Figure 10 change if frequency of precipitation is used?

ANSWER: We thank the referee for the very good suggestion of using precipitation frequency. In the new version of the manuscript we have substituted the precipitation rate with the daily precipitation frequency. This variable is constructed as follows: for each grid point, if it rains during a specific day, we set the value to 1, and to 0 if it doesn't rain. With this choice, we obtain results that we believe correspond to a clearer narrative. As a result, we have radically altered the section of the manuscript dealing with the rainfall analysis.

QUERY: I also think that it is adequate that the authors include at least one comment on the fact that considering all grid boxes in the Northern Hemisphere might be "masking" the predictability and/or the dynamical properties of the variables in study, compared to performing the analysis in regions that are known to have a more homogeneous climatic response to different sources of predictability. In other words: how different are the results if instead of considering the entire Northern Hemisphere, regions of homogenous predictability are considered? Seasonality is different for different regions, so is it possible that mixing the seasonal cycle of so many places degrades the characterization of the dynamical properties of the system?

ANSWER: In a recent paper (Faranda et al. 2017, Sci. Rep.), we analyzed a specific region of the northern hemisphere, i.e. the North Atlantic, which is more homogeneous in terms of predictability. Indeed the results obtained at the regional scale do not always

match those obtained for the full Northern Hemisphere. However, rather than ascribing this discrepancy to a "masking" we rather interpret it as meaning that, despite the large regional differences, our methodology still allows us to draw interesting conclusions about the full hemispheric dynamics. Of course, other studies on restricted regions may answer more specific questions about the predictability and the persistence of certain patterns. Our study remains mostly theoretical at this stage and focuses on the general properties of the atmospheric attractor. We have clarified this in the conclusions of the manuscript.

QUERY: P2L12: excise "the" before "attractor"

ANSWER: Corrected

QUERY: P2L33: how many are "enough"?

ANSWER: We have now specified this.

QUERY: P2L34-P3L1: something is wrong. There is a "." misplaced, and I think the trajectory is around $\xi$, not around $\theta(\xi)$, which has not been defined yet.

ANSWER: We have corrected these typos.

QUERY: P3L7: I suggest to change "air" by something else. Several of the readers of the paper are going to be fast-readers, and in several other places "air" might be misleading. Perhaps use "t2m", which is one of the standards.

ANSWER: We have replaced "air" by "t2m" as suggested.

QUERY: P3L8: what do the authors mean by "peculiar subspace"? Clarify.

ANSWER: We have now explained this terminology.

QUERY: P5L2: further explain what the "length" of a cluster is.

ANSWER: This is now explained.

QUERY: P5L15-16: excise the subjective phrasing "a very".

ANSWER: Done.

QUERY: P5L22: maybe the semi-annual cycle is inherited from some semi-annual variability phenomena too? e.g., North Atlantic Sub-tropical High?

ANSWER: Good suggestion, we have emphasized this in the discussion of the slp results.

QUERY: P5L28: why is it more restrictive?

ANSWER: Because if one considers the typical partition into 4 weather regimes, the probability of being in one of them is roughly 1/4=0.25, whereas the probability of being close to zeta is set by the quantile: for q=0.98, the probability of being close to zeta is 0.02. This is now better explained in the text.

QUERY: P6L16-17: it seems to me that the weakened Aleutian low is actually a high in that figure. Can you comment on that? Also, the magnitudes of the anomalies are really low (amplitude of 10 hPa). This might be associated, again, with the fact that all days are considered together, without seasonal discrimination. The Aleutian low is typical of winter; can the authors comment on why it should be expected to appear during the maxima of d

ANSWER: The plots show anomalies relative to a climatological seasonal cycle, so that a positive anomaly over a climatological low would correspond to a weakened low (provided that the anomaly is not so strong as to turn a low into a high). However, following the Reviewer's first major comment we have radically altered the presentation and discussion of the geographical composites, so that the figure mentioned here is not present in the revised version of the manuscript.

QUERY: P6L26: please explain why that means intra-seasonal correlation.

ANSWER: We have removed this sentence and now discuss the cross-correlation only in reference to Figure 4b.

QUERY: P7L11: any hypothesis for why is it absent?

ANSWER: This is linked to the slight offset in the minima of the seasonal cycle of d and theta, which result in a positive ACF for d between winter and summer and a negative one for theta. We illustrate this in Fig. R1 below, which shows the d and theta curves for t2m shifted by 6 months. We now briefly mention this in the text.Fig. R1: Local dimension (continuous lines) and theta (dashed lines) for t2m. The blue lines are lagged by 6 months relative to the red lines.

QUERY: P7L22: Explain what you mean by "high-dimensional situation".

ANSWER: Here we meant a situation with a high d value, and hence a high number of active degrees of freedom, which makes the evolution of the atmospheric state inherently hard to forecast. We have now rephrased this to clarify our point.

QUERY: P7L30: it is a bit strange that extremes cannot be matched to large-scale patterns, as they tend to be controlled by synoptic configurations. Maybe what the authors mean is that there is no global pattern associated with the extremes? I do not expect that case either. Right now, the sentence is confusing.

ANSWER: We have now repeated the analysis with a different variable (precipitation frequency) and this sentence has been removed as this is not the case anymore.

QUERY: P8L12: "Precipitation rate"

ANSWER: We have now corrected this to "Precipitation Frequency" to reflect our updated analysis. QUERY: P11L9-10: use "twenty" and "fifty", or "20" and "50".

ANSWER: Format of numbers is now consistent.

QUERY: P11L16: along the paper, the authors use 1000 hPa. Did they interpolate? Or do they just mean "surface". If the latter, please change to "surface".

ANSWER: 1000 hPa has been changed to surface.
QUERY: P11L26-27: I actually expect to see a correlation between pressure gradients and monsoonal precipitation. Maybe I'm not understanding something here.

ANSWER: We have now included a revised explanation in this section discussing the new precipitation frequency results.

[Figure]

[Figure]

**Fig. 1.** Local dimension (continuous lines) and theta (dashed lines) for t2m. The blue lines are lagged by 6 months relative to the red lines.

---

## Author Comment (AC3) · 31 Oct 2017

In the last answer, we actually refer to Figure 12 instead of Figure 13

---

## Author Comment (AC4) · 31 Oct 2017

In the answer to second query, "These are shown in the new Fig 3.' should be "These are shown in the new Figs 5-8-11".
* * *

---

## Author Response (AR1)

**Dear Editor,**

**We provide a detailed answer to the referees' comments below. We hope that the new version is now suitable for publication in NPG.**

**Best Regards,**

**The authors**

**Referee #1**

Authors present an important application of the of the extreme value theory to Poincaré recurrences in dynamical systems, which can provide estimations of some metrics of the ergodic chaotic attractors, namely the phase-space local dimension and local recurrence. Authors apply the technique to relevant daily fields: the sea level pressure, surface air temperature and precipitation rate, trying to understand their seasonal dependency in the Northern Hemisphere and exploring the atmospheric patterns which are consistent with maximum and minimum values of those metrics. There are, however some minor worthwhile points which should be addressed before publication.

**We thank the Referee for the positive comments. We provide a point-by-point reply below.**

1 – Annual cycles of the local dimension d for slp and Temp(air) are in phase opposition (Figs. 2a, 2c). There is any physical reasonable explanation for that? The authors present a hypothetical explanation for the minimum of d for slp at the summer and maximum at the equinoxes as being more 'unstable' because of an incremented exploration of the summer and winter configurations. However, that could apparently be valid for any field but in fact it is totally contradicted in the case of the Temp(air) where d is higher in summer. Therefore, the authors should reformulate the explanation (which, as presented looks somehow ad-hoc) or give and explanation why it does not work for Temp(air).

**This is an interesting question that motivated some further discussion. Eventually, we concluded that the t2m dynamics fits into a single potential well dynamics, with the extremes located in winter and in summer. We can imagine a sort of Langevin model for slp-t2m, where the slp is the variable pushed in the "winter" or "summer" potential wells, while the temperature acts as forcing noise term (i.e. single potential well dynamics) with extremes in summer and winter. We have now added this possible explanation to Sect. 3.2 of the manuscript.**

2 – On pg. 8 in the Analysis of Precipitation data. Authors explain the higher value of d (compared to slp and air) thanks to the scattered and noisier character of the precipitation field. However, analysis of precipitation extremes has been done using the multifractal analysis (MFA) (Langousis et al. 2009) which also give hints about the dimension. Authors should try to relate the presented results with those of MFA.

**We have read with interest the paper suggested by the Referee and included it in the discussion of prp results. We have also replaced the precipitation rate with the precipitation frequency, as suggested by Referee #2. This has led to a radical restructuring of the section analyzing the precipitation data.**

3 – The graphs of ACFs(s) (Fig.1 and others) present discontinuities or overlaps near the zero line. This is maybe because of the symbol size? Use a format avoiding that artifact.

**Thanks for spotting this, the original ACF figures have been replaced with ones displaying the values with cross-shaped markers to avoid artefacts.**

4 – Caption of Fig. 11 should be more complete, i.e. making explicit the pairs of variables: Fig11a,b (slp-air), Fig. 11c,d (slp-prp), Fig. 11e,f (air-prp). Which variable lags in the future?

**We have updated the panels in that figure that currently is figure 12, and we have explained which variables lag in the future.**

LANGOUSIS, A., VENEZIANO, D., FURCOLO, P.Y. LEPORE, C. (2009): Multifractal rainfall extremes: Theoretical analysis and practical estimations. *Chaos, Solitons and Fractals*, n° 39, 1182-1194.

**Referee #2**

The authors analyze dynamical properties of daily surface pressure, surface temperature and precipitation rate using two parameters from a universal distribution derived in previous research from extreme value theory of Poincaré recurrences. The two parameters, local dimension of the phase space attractor and the inverse of the average residence time, are physically intuitive, and have the potential to characterize predictability of the analyzed variables. The manuscript is of interest for both the Dynamical System and Applied Climatology scientific communities, illustrating an approach that could be used in other regions of the world and for other variables of interest. Nonetheless, the analysis raises three main issues that I think need to be clarified before the paper can be considered for publication in NLPG. In addition, a number of minor changes/clarifications are also suggested to the authors.

**We thank the referee for the detailed comments, which we have addressed as detailed below.**

**Main issues**

The manuscript identifies seasonality as a key regulator of the behavior of local dimension and persistence. For example, for surface pressure, two maxima and minima per year are reported, in seasons considerably different (e.g., the typical associated teleconnections are not the same in the two max seasons, with potential implications for the phase-space trajectories and their characteristics). Nonetheless, my understanding is that the composite of anomalies shown in Figure 4 for the maxima and minima are computed from the "extreme" regions in the dimension/persistence diagram (Figure 3, in the example considered here), without discriminating between the two seasons. As a result, I wonder if the present composite analysis

is not averaging fields that correspond to different situations. Please clarify this point and the possible differences that could be obtained if analyzing separately the two different maxima and minima.

**The referee is right about the seasonal dependence of the results. We have now added a new Figure 3 showing the distribution of dynamical extremes for every season. We note that in most cases the bulk of the maxima or the minima of the dimension/persistence corresponds to a single season. This is however not true for SLP, which shows a greater spread between seasons. For all the cases where two or more seasons have > 25% of the extremes, we have therefore produced separate composites for each season. These are shown in the new Fig 5-8-11. We have further added a sign test to all geographical composites to display the regions where more than 2/3 of the composite members agree on the sign of the anomalies. We have extensively modified the manuscript in Section 3 to account for the updated analysis and the new figures.**

My second major concern is somehow related to the first one. Although something similar might be happening for precipitation rate (there is only one maximum, but it is so long that different climate agents might be involved, and thus perhaps it makes sense to try to sub-sample that season), I wonder if less noisier results could be obtained if a different precipitation variable is used. As the authors are probably aware, precipitation rate is in general too noisy and far less predictable than accumulated rainfall or rainfall frequency, the latter being the most predictable of all three – at least in most parts of the world. How much would Figure 10 change if frequency of precipitation is used?

**We thank the referee for the very good suggestion of using precipitation frequency. In the new version of the manuscript we have substituted the precipitation rate with the daily precipitation frequency. This variable is constructed as follows: for each grid point, if it rains during a specific day, we set the value to 1, and to 0 if it doesn't rain. With this choice, we obtain results that we believe correspond to a clearer narrative. As a result, we have radically altered the section of the manuscript dealing with the rainfall analysis.**

I also think that it is adequate that the authors include at least one comment on the fact that considering all grid boxes in the Northern Hemisphere might be "masking" the predictability and/or the dynamical properties of the variables in study, compared to performing the analysis in regions that are known to have a more homogeneous climatic response to different sources of predictability. In other words: how different are the results if instead of considering the entire Northern Hemisphere, regions of homogenous predictability are considered? Seasonality is different for different regions, so is it possible that mixing the seasonal cycle of so many places degrades the characterization of the dynamical properties of the system?

**In a recent paper (Faranda *et al.* 2017, *Sci. Rep.*), we analyzed a specific region of the northern hemisphere, i.e. the North Atlantic, which is more homogeneous in terms of predictability. Indeed the results obtained at the regional scale do not always match those obtained for the full Northern Hemisphere. However, rather than ascribing this discrepancy to a "masking" we rather interpret it as meaning that, despite the large regional differences, our methodology still allows us to draw interesting conclusions about the full hemispheric dynamics. Of course, other studies on restricted regions may answer more specific questions about the predictability and the persistence of certain patterns.**

**Our study remains mostly theoretical at this stage and focuses on the general properties of the atmospheric attractor. We have clarified this in the conclusions of the manuscript.**

**Minor details**

P2L12: excise "the" before "attractor"

**Corrected**

P2L33: how many are "enough"?

**We have now specified this.**

P2L34-P3L1: something is wrong. There is a "." misplaced, and I think the trajectory is around $\xi$, not around $\theta(\xi)$, which has not been defined yet.

**We have corrected these typos.**

P3L7: I suggest to change "air" by something else. Several of the readers of the paper are going to be fast-readers, and in several other places "air" might be misleading. Perhaps use "t2m", which is one of the standards.

**We have replaced "air" by "t2m" as suggested.**

P3L8: what do the authors mean by "peculiar subspace"? Clarify.

**We have now explained this terminology.**

P5L2: further explain what the "length" of a cluster is.

**This is now explained.**

P5L15-16: excise the subjective phrasing "a very".

**Done.**

P5L22: maybe the semi-annual cycle is inherited from some semi-annual variability phenomena too? e.g., North Atlantic Sub-tropical High?

**Good suggestion, we have emphasized this in the discussion of the slp results.**

P5L28: why is it more restrictive?

**Because if one considers the typical partition into 4 weather regimes, the probability of being in one of them is roughly 1/4=0.25, whereas the probability of being close to zeta is set by the quantile: for q=0.98, the probability of being close to zeta is 0.02. This is now better explained in the text.**

P6L16-17: it seems to me that the weakened Aleutian low is actually a high in that figure. Can you comment on that? Also, the magnitudes of the anomalies are really low (amplitude of 10 hPa). This might be associated, again, with the fact that all days are considered together, without seasonal discrimination. The Aleutian low is typical of winter; can the authors comment on why it should be expected to appear during the maxima of d

**The plots show anomalies relative to a climatological seasonal cycle, so that a positive anomaly over a climatological low would correspond to a weakened low (provided that the anomaly is not so strong as to turn a low into a high). However, following the Reviewer's first major comment we have radically altered the presentation and discussion of the geographical composites, so that the figure mentioned here is not present in the revised version of the manuscript.**

P6L26: please explain why that means intra-seasonal correlation.

**We have removed this sentence and now discuss the cross-correlation only in reference to Figure 4b.**

P7L11: any hypothesis for why is it absent?

**This is linked to the slight offset in the minima of the seasonal cycle of d and theta, which result in a positive ACF for d between winter and summer and a negative one for theta. We illustrate this in Fig. R1 below, which shows the d and theta curves for t2m shifted by 6 months. We now briefly mention this in the text.**

[Figure]

*Fig. R1: Local dimension (continuous lines) and theta (dashed lines) for t2m. The blue lines are lagged by 6 months relative to the red lines.*

P7L22: Explain what you mean by "high-dimensional situation".

**Here we meant a situation with a high d value, and hence a high number of active degrees of freedom, which makes the evolution of the atmospheric state inherently hard to forecast. We have now rephrased this to clarify our point.**

P7L30: it is a bit strange that extremes cannot be matched to large-scale patterns, as they tend to be controlled by synoptic configurations. Maybe what the authors mean is that there is no global pattern associated with the extremes? I do not expect that case either. Right now, the sentence is confusing.

**We have now repeated the analysis with a different variable (precipitation frequency) and this sentence has been removed as this is not the case anymore.**

P8L12: "Precipitation rate"

**We have now corrected this to "Precipitation Frequency" to reflect our updated analysis.**

P11L9-10: use "twenty" and "fifty", or "20" and "50".

**Format of numbers is now consistent.**

P11L16: along the paper, the authors use 1000 hPa. Did they interpolate? Or do they just mean "surface". If the latter, please change to "surface".

**1000 hPa has been changed to surface.**

P11L26-27: I actually expect to see a correlation between pressure gradients and monsoonal precipitation. Maybe I'm not understanding something here.

**We have now included a revised explanation in this section discussing the new precipitation frequency results.**